# Winning against All Odds: Elly Schlein's Successful Election Campaign and Instagram Communication Strategies

**Daniele Battista**

Departments of Political and Social Studies, University of Salerno, 84084 Fisciano, Italy; dbattista@unisa.it

**Abstract:** The article aims to examine the characteristics of mediatized political communication in a hybrid media system, in which the social media accounts of political leaders play an increasingly important role. Firstly, the phenomenon of mediatization and its main aspects in relation to the mass media system are described. Subsequently, the concepts of disintermediation and new forms of mediation specific to social platforms are discussed. The analysis shows that disintermediation does not lead to the end of mediatization, but emphasizes the need to support the mass media logic based on the intertwining of entertainment and a social logic built on online content capable of triggering the creative potential of digital media. In particular, it is highlighted how politicians now prioritize social media over traditional media, assuming the role of social opinion leaders. The analysis of the posts published on the "ellyesse" Instagram account in the period from 4 December 2022, the date of the announced candidacy for the party secretary, to 26 February 2023, the day of the primaries in which he won against all odds, reveals a tendency towards the remediation of traditional content accompanied by innovative use of digital images, aimed at creating a sense of immediacy in interaction with the public. Furthermore, the use of codes typical of mass media mediatization is observed, but in a context where they change their meaning and assume a new communicative function. All these aspects reinforce the idea that social media, in a relationship of complementarity and interdependence with mass media, orient political communication towards the selection and repetition of a coherent set of identity traits.

**Keywords:** political communication; pop politics; electoral campaigns; lifestyle politics; intimate politics

## 1. Introduction

The Census report highlights a strong increase in internet usage by Italians between 2021 and 2022, reaching 88.0% of users, with an increase of 4.5% compared to the past. Furthermore, 88.0% of users use smartphones, with an increase of 4.7% compared to the past. Finally, the number of social network users has increased by 5.8%, reaching a total of 82.4%. In truth, democracies around the world are undergoing a major transformation in communication systems that also affect the political sphere. It seems like a distant era when the arrival of television brought a wave of changes to the political scene after World War II. And even the times when politicians were reluctant to join and land on various social media platforms seem very distant (Bentivegna 2014). Undoubtedly, the passage of time has led to a change in the tools used for political communication. Currently, it can be argued that political leaders have identified in Instagram an opportunity to access a new target audience, characterized by a younger age and a lesser familiarity with the political sphere (Sampietro and Sánchez-Castillo 2020). This audience, in fact, predominantly uses the Instagram platform for recreational purposes and to interact with other users, rather than to discuss political issues or to acquire public information (Manikonda et al. 2016). The aforementioned data shows that politics must increasingly confront the digital transformation of the media system. The central issue is to understand how social media culture is influencing politicians in terms of storytelling, codes, and communicative behaviors. The processual and open nature of this question implies that

a definitive answer cannot be provided, as each politician can interpret their presence on social networks differently. However, it is important to recognize that the relationship between social media and politics is not purely idiosyncratic, but rather connected to the media culture that the politician contributes to shaping and making perspicuous. Moreover, this relationship does not arise out of nowhere, but is closely related to the long and complex history of the relationship between mass media and politics. In this sense, the effectiveness of the individual politician's action depends on their ability to coherently integrate themselves within a media culture that is progressively developing. The presence of social media in the landscape of political communication requires a complex analysis that considers the intersections, continuities, and differences with the still-dominant pop-television model. In order to account for this complexity, the article is divided into two parts. In the first part, classical theoretical concepts used for the study of political communication are introduced in order to suggest a redefinition in light of the disintermediation operated by digital media and, in particular, social networks. However, such concepts will not be analyzed in an abstract way, but in the concreteness of the hybrid media system (Chadwick 2013), in which social networks can define their specificity through their relationship with other media. The second part of the work will focus on the qualitative analysis of Elly Schlein's Instagram account, aimed at examining how certain aspects of mass political communication have been amplified, redefined, and partially used thanks to the emerging new logic of social media. While not exhausting all political uses of social media, this case study packaged around the new Secretary of the Democratic Party represents a significant and promising example regarding the broader process of mediatization of politics in Italy on several fronts.

- **Who is Elly Schlein? The political history and vision of the new Secretary of the Partito Democratico.**

The change in leadership in the race for the Secretary of the Partito Democratico was unexpected in the various predictions of the eve and probably represents the most significant event in Italian politics in the first quarter of 2023. The political figure, who was external to the party until a few days before the presentation of the candidacy, surprisingly defeated the party establishment, which had heavily bet on Stefano Bonaccini as the representative of continuity in the footsteps of the work of the previous secretaries. Although Bonaccini had overwhelmingly won the first round, reserved only for PD members, with 54% of the votes against Schlein's 36%, the result was completely overturned in the second round, open to all Italian voters over 18, with 54% of the votes in favor of Schlein and 46% in favor of Bonaccini. This result was largely determined by the contribution of those who were not registered with the party. Elly Schlein, Member of the Italian Parliament, former Vice President of the Emilia–Romagna Region with responsibilities for combating inequalities and ecological transition, with her victory can represent a significant surprise and opens up interesting new scenarios worth exploring. But her commitment to politics goes way back; in fact, she participated in various mobilizations in support of Barack Obama's presidential candidacy in the United States. In Italy, she launched the Occupy PD mobilization campaign in 2013, which started from the youth base of the Democratic Party, regarding the choice of national leaders to form the Letta government, which led to the occupation of some party headquarters. Schlein supported Pippo Civati in the 2013 PD primaries and subsequently was elected to the party's national direction. In 2014, she ran for the European elections with the Democrats, where she was elected with over 53,000 preferences and served as vice president of the EU–Albania Stabilization Parliamentary Commission delegation, as well as a member of the Development Committee. In 2015, Schlein left the party to join the Possibile party founded by Civati. In 2020, she was elected Vice President of the Emilia–Romagna Region with the Brave Ecologist and Progressive Emilia–Romagna list, subsequently becoming Bonaccini's deputy. Schlein was the most voted list candidate in the history of Emilia–Romagna with almost 23,000 preferences. In the 2021 political elections, she was nominated as an independent in the PD list for the Chamber of Deputies, where she was subsequently elected, resigning from the position of vice president of the

Region. Only after Enrico Letta's resignation in November 2022, Schlein announced her candidacy for the PD secretaryship through an Instagram live stream. Her program promotes social equity and environmental protection, considering them closely connected. Among the main points are the fight against fragilities, the safeguarding of social and civil rights, the defense of the public health service, fiscal reform and taxation of large assets, the right to housing and the relaunch of public residential construction, as well as the modification of Law 194 on voluntary termination of pregnancy through the guarantee of a percentage of non-objecting doctors. Schlein emphasizes the importance of not ranking social and civil rights and insists on the incompatibility between party positions and public and administrative functions. Her goal is to reform the Democratic Party and create a clear identity to attract new supporters. Additionally, Schlein has been supported by numerous famous figures, including the singer Levante and the director Gabriele Muccino, who have supported her for her political authenticity and left-wing position.

## 2. From Traditional Politics to Pop Politics: The Phenomenon of Mediatization and Popularization of Politics through Mass Media

Politics, in representative democracies, is inseparable from a process of public, informal, and permanent communication through which the political freedom of citizens can be exercised even beyond the moment of voting. This process accompanies and supports the deliberative procedures of institutions, giving rise to the democratic diarchy of will and opinion (Urbinati 2014). In contemporary democracies, this is mainly achieved in a mediated public space (Mazzoleni and Schulz 1999), shaped by technologies and practices that can make the set of issues considered of public interest a common experience. It follows that both the institutional dialectic, aimed at deliberation, and the informal one, aimed at information and consensus-building, must enter into some relation with a reference media system (Esser and Stromback 2014). This relationship is thus a structural element of democracies, capable of characterizing, according to its specific determinations, both the decision-making process and that of consensus formation (Blumer and Kavanagh 1999). It is important to clarify, however, that when referring to the media system in relation to the political system, we are not only referring to a set of technologies. After all, the mass media are themselves constituted as distinct institutions (Hjarvard 2008), in which specific professional elites determine the possibilities of the technological medium according to their own set of criteria and objectives. By tracing the reconstruction of Altheide and Snow (1979), it is the media elites who establish the formats of their respective media, through which the communicative flow is concretely structured. This generates a culture of media production and reception in which formats function as shared communicative norms, which contribute to selecting what is significant and establish the way in which meaning shapes the space and time. From the analysis, it emerges that formats influence communication from both the content and form perspective. The format is the organizational structure of the material, the style of presentation, the focus on specific behavioural characteristics, and the grammar of mediated communication. Moreover, it becomes the framework or perspective used to represent and interpret phenomena. Media institutions, driven by the pursuit of profit through audience maximization, constitute the prerequisites of the democratic public sphere. After all, we are all immersed in what is defined as a plebiscite of the audience (Urbinati 2012). It follows that media institutions pursue purposes that differ from politics, based on the logic of spectacle or entertainment to ensure audience attention. These institutions seek to satisfy the tastes and consumption habits that they themselves have contributed to creating; and, therefore, must reconcile professional aspects with a broader and pervasive commercial logic. There is also an additional element to consider, namely that the increase in cultural influence deriving from media companies has led to a progressive mediatization of politics (Strömbäck 2008). Politics has become increasingly mediatized, both through the communicative actions of its actors and through the narration provided by the media, which tend to conform to the formats and logic of the reference media system. This results in a spectacularization of public debate with phenomena such

as personalization, simplification, polarization, and the representation of elections as an obstacle course in the logic of being fully immersed in what is defined as a permanent election campaign (Blumenthal 1980); or a campaign without rest or limits, and which is no longer based on temporal logic (Sorice 2011). At the same time, politicians adhere to this logic to obtain greater visibility, influence over the media agenda, and greater proximity to citizen-viewers. From a linguistic point of view, the mediatized politician abandons the technical language of politics (Dell'Anna 2010) in order to speak a language that voters are already accustomed to from media consumption. In the context of the mediatization of politics, there is a strong intertwining of the logics and objectives of the media with those of the political sphere, in a sort of reciprocal symbiosis (Splendore 2014). This phenomenon is often referred to as the popularization of politics, emphasizing how public discourse tends to conform, within the new digital space led by the media, to the standards of entertainment and the mass audience's imaginary (Mazzoleni and Sfardini 2009). In the contemporary dynamics of political mediatization, the political leader must assume a triple role: that of institutional representative, spokesperson for the political party, and media public figure. In this context, a transversal operation takes place aimed at making political content popular and politically exploiting popular content. This latter aspect takes various forms, including celebrity politics (Street 2004; Wheeler 2013; Campus 2020), intimate politics (Stanyer 2012; Mazzoni and Ciaglia 2015), and lifestyle politics (Bennett 1998). In particular, these concepts refer to the tendency of political leaders to adopt communication strategies based on themes and images that reflect the interests, values, and lifestyle of ordinary citizens, in order to strengthen their media presence and political influence. Everything related to the private sphere of the political star, from their love life to their soccer fandom, is allowed onto the stage of the mediatized public sphere (Ceccobelli 2017). All these elements are exploited by politicians for their potential representativeness and to reach groups of citizens not directly interested in politics. This means that these elements are brought to the forefront and publicly discussed, playing a significant role in the construction of the politician's image and their relationship with the public. In this way, politicians seek to use their private life as a connection element with the public, showing aspects of their personality and daily life that can arouse empathy and interest among voters. With their personality, they respond to the overall trend of showcasing private life (Codeluppi 2007), that is, to privilege the visibility of the private sphere. The privatization of politics marks the shift of importance from television to social media and decrees a fluctuating public that is less interested in administrative qualities and more in personal ones, or rather, in the characteristics that bring the politician closer to people in their daily lives. This also corresponds to the progressive downsizing of citizens' formal political engagement, who direct their interest more towards the politician's person than towards the organization and political programs (Rega and Bracciale 2018). It is essential to underline that the use of "pop" communication codes and styles is not capable of providing a judgment on the level of democratic quality of the mediatized public space. In fact, mediatization is not a normative concept but rather a description of a social phenomenon that occurs within contemporary societies. This means that, while the adoption of "pop" communication strategies can lead to the trivialization of politics, on the other hand, it can also generate a more inclusive and open public space for the participation of different voices and perspectives. In this sense, mediatization can offer new opportunities for access and interaction between citizens and political leaders, promoting the creation of a more democratic and pluralistic public sphere. However, it should be noted that this process does not happen automatically and depends on the ability of institutions and civil society to promote and protect a culture of participation and dialogue. Moreover, it is important to consider that the democratic quality of the public space is not limited to the communicative dimension but also depends on the ability to ensure effective representation and participation of citizens in political decisions.

### 3. The Evolution of the Media System That Became Hybrid: From the Diversification of Logics to the Impact of Disintermediation on Social Media

Despite networked politics (Cepernich 2017) surpassing television in terms of products, according to consumption data in Italy, there is a stable trend in the general consumption of television content. This stability is determined by the contraction of the number of viewers of traditional television, digital terrestrial, which has decreased by 3.9% compared to the previous year. At the same time, there is a slight increase in the consumption of satellite television, which is 1.4%. However, there is a significant increase in the consumption of television via the internet, with a user base of 52.8% of the population, or more than half of Italians, with a growth of 10.9% in just one year. In particular, mobile TV has registered exponential growth, going from 1.0% of viewers in 2007 to 34.0% today, which represents more than a third of Italians. Although from this overview, television continues to play a central role in the media diet of Italian citizens, it is becoming increasingly evident how the public space is constituted through a hybrid media system in which digital media assumes increasing importance. However, it should be emphasized that digital and traditional media are in constant dialectic of competition and interdependence according to a movement of continuous fragmentation and reintegration of the communicative flow as supported by Chadwick (2013). While there is indeed a great qualitative and quantitative differentiation of media channels and formats, none of these is independent of the others. A concrete example of the interaction between digital and traditional media is the need to reach a wide audience. To capture the attention of a broad audience, content produced on digital media must be synthesized and disseminated through traditional means of communication, such as television and radio. Similarly, to achieve lasting and significant impact, the content broadcast by traditional media must be discussed and amplified online through digital media channels. This interaction between digital and traditional media allows content to reach a wider and more diversified audience, thereby increasing its potential to influence public opinion and promote debates and conversations on society and politics. However, this process also requires some attention from content producers, who must adapt their communication to the specific characteristics of different means of communication and their target audiences. More generally, it emerges that Web tools and current communication strategies are advancing in this direction, proposing a system of intercreativity (Fernández-Castrillo 2014). In fact, the continuity between offline and digital dimensions is now consolidated, in which citizens move and act to express opinions and express their own feelings in a social climate of growing media personalization, in line with a multidimensional dimension of life that unfolds between online and offline (Boccia Artieri 2012). The new environment is determined by networks well before the development of social platforms (Boccia Artieri and Marinelli 2018); and with the diffusion of new digital technologies, the development of the social environment poses a new ecology of social and political relationships between groups and individuals. The communication ecosystem is reinforced by interactive media, and the debate in the extended public sphere is without intermediaries, and involves users who transcend geographical and cultural barriers (Van Dijck et al. 2018). This interaction demonstrates how digital and traditional media are complementary to each other in reaching a wide and diversified audience. Political leaders' communication can, thus, increasingly approach a transmedia narrative (Jenkins 2006). In this illustration, public debate takes the form of a repetitive story that unfolds across multiple media platforms, each of which provides a unique contribution and selects its own group of topics, actors, and target audience, using different appropriateness standards. The specific modes of interaction between media are determined mainly by the characteristic of Web 2.0; i.e., "disintermediation," which allows participants to act as independent editors, ignoring the need for traditional intermediaries and central control over the flow of information (Bentivegna 2015; Giacomini 2018). Social media allow for a direct and horizontal form of communication without intermediaries between sender and receiver, with the potential for anyone to become a sender or receiver, without limits on topics or modes of expression. This differs from the traditional media system, characterized by

formats produced by mass media elites and transmitted vertically to a passive audience. The theory of mediatization argues that media have become the main channel through which politics is presented, interpreted, and judged. In contrast, disintermediation suggests that the elimination of intermediaries, in this case, media, could limit the influence of media logic. This could allow politicians to act without having to satisfy professional standards imposed by the media and to relate directly to citizens in their role as political representatives. In other words, disintermediation could represent a significant change from the theory of mediatization, as it could allow politicians to dissociate themselves from the roles of media figures and establish a more authentic relationship with citizens. This could be beneficial in terms of increased trust and engagement of citizens in politics, but could also pose risks and challenges for political communication and the construction of politicians' public image.

It can be argued that the use of social media by politicians perpetuates mediatization, both through the confirmation of established media forms and through the creation of new ones. This happens because the use of social media by politicians continues to increase, confirming the role of media in politics and increasing the importance of the media dimension in political communication. Furthermore, social media have opened up new possibilities for political communication, such as direct and unfiltered communication with citizens, user-generated content creation, participation in public conversation, and the use of personalized political marketing techniques. These elements further amplify the importance of media in politics, not only as communication channels, but also as tools to influence public opinion and citizen political participation.

The first reason why the use of social media by politicians contributes to mediatization is related to the continuous interaction and integration of media systems. In fact, for a social media content to be transmitted by traditional media, it must adapt to the standards of consolidated media formats. Moreover, the codes of traditional media remain of great importance in the imagination and common competence. This is demonstrated by the fact that Beppe Grillo, the Italian politician who made the web his flag, was also a celebrity of political entertainment television. Looking outside the country, the case of Zelensky is emblematic, a former comedian and television actor who was elected president of Ukraine in 2019 with over 70% of the votes in the second round of the presidential elections, leading his party named after the television series in which he starred. The current Ukrainian president was known for his portrayal of the character Vasyl Holoborodko in the television series "Servant of the People," in which a history teacher becomes president thanks to his fight against corruption. This suggests that the influence of traditional media on the way politics is communicated and represented continues to be significant, despite the increasing importance of social media as a channel for political communication. Another reason stems from the concept that disintermediation is actually produced by social media platforms. In fact, behind the appearance of immediacy-horizontalism, new and different forms of verticality-mediation are reintroduced, in a sort of combination within the framework of a neo-intermediation. Each social platform implicitly intervenes in two types of mediation: on the one hand, it rigidly delimits the linguistic options available to users within which they can act, for example, by imposing time limits or maximum length for content; on the other hand, it algorithmically intervenes by selecting and organizing both the posts that appear in the feed, as well as the recommendations of new profiles to follow and advertising content. All of this constitutes a set of strategic choices made by a circle of subjects who act as intermediaries between the sender and the recipient of communication. The algorithm itself operates effectively only based on the values dictated by those who define its criteria. This set of choices also follows a very specific purpose and logic, specific to social networks and aimed at profiling personal identities as a set of activities, interests, and consumption habits (Floridi 2014). The decision to follow a politician on social networks can contribute to collecting data on the personal characteristics of the user, thus fueling the profiling process. The processing of this data through algorithms allows for content to be adapted to the user's tastes and preferences, creating a statistically optimal sort of personalization.

However, in a completely parallel way, with the increasing emphasis on techno-social phenomena such as echo chambers (Sunstein 2017) and filter bubbles (Pariser 2011), we are brought to relate only to individuals and informative sources with whom we share some traits of similarity. All of this stems from the acceptance and more or less conscious practice of this media logic by the users of the platforms (Van Dijck and Poell 2013; Klinger and Svensson 2015). Despite everything, a third form of re-intermediation is closely linked; in fact, although they have internalized their condition as prosumers (Boccia Artieri 2017), not all social media users enjoy the same communicative strength. The platforms are indeed articulated in asymmetric–vertical relationships, which partly reproduce offline power relationships, in which a few accounts are followed, commented on, and shared by many. This further and new intermediation revitalizes, modifying its meaning, the notion of opinion leader provided by Lazarsfeld (1944). The choice to follow specific accounts on social networks contributes significantly to the construction of users' informational identity, whose online profile is influenced by the type of content and informative sources they follow. Opinion leaders, exploiting their supposed authority and creating a relationship of trust with followers, carry out a simplified selection of information in various areas of interest, including politics. This raises the question of how the media logic of social networks connects with political communication. Unlike the strategy of reaching less politically interested citizens through entertainment, the choice to follow a politician on a social platform requires a level of active involvement and cannot be considered a simple selfie on Facebook. On the other hand, despite the selfie being among the most popular forms of communication with which popularity increases and media attention is drawn (Karadimitriou and Veneti 2016), it still represents an act of technological rationality (Marcuse 2002) that does not shift the balance. On social media, therefore, the citizen's interest is the premise, not necessarily the result of the communicative process. Following a politician should be thought of as part of a broader process of defining one's spheres of interest. Moreover, if this logic is adopted by politicians, one could speak of a new kind of mediatization that complements and accompanies mass media mediatization. Alongside the "political star", there would be a "political opinion leader", whose communication is aimed at creating and constantly involving a community identified by a set of strong and clearly recognizable identity characteristics. This media logic of identity is recognizable in the use that Italian politicians are already making of social media. How does it intertwine with pre-existing mass media mediatization? Through what modes of actualization of the communicative possibilities of their formats? To answer these questions, we analyzed the set of posts (245) published by Elly Schlein through her Instagram account @ellyesse from 4 December 2022 to 26 March 2023. The selected political leader was chosen based on her activity on social media, which has demonstrated significant engagement from her community of followers, despite the number of followers being lower compared to the other competitor for the party leadership and to several national political leaders in general. Despite Elly Schlein's strong presence on social media, her number of followers is still much lower compared to the leaders of other political parties, such as Matteo Salvini and Giorgia Meloni. However, the choice to analyze her campaign was made because the strategy adopted during the Democratic Party primaries focused on attracting the attention of the entire democratic world, not just internal party members, by building a strong and autonomous identity that rejected any form of political and cultural subordination. Schlein also addressed a series of issues that attracted the attention of a wide range of voters, including those of the Five Star Movement, More Europe, and Left and Greens, through clear and concise communication that helped position her candidacy as a future leader of the entire progressive world. This is the main reason that contributed to advancing work on her figure, which will most likely be central in the coming years in the chaotic Italian political dynamics. In particular, the focus was on the Instagram platform, as it has shown to generate a high level of interaction and rapid growth. The analysis will not be limited to showing the social communication of a single political actor, but aims to illustrate the trends and issues of the process of social mediatization of Italian politics. The new secretary

of the Democratic Party, in this regard, represents a significant example of an ongoing phenomenon that, although still evolving, seems to indicate some general trends. The objective of this study is, therefore, to deepen the nature and effects of the mediatization of Italian politics through the analysis of this political leader's social activity.

## 4. Re-Mediation, Modifying Interventions, and Speed as an Influential Factor in Creating Visual Content on Digital Platforms

The corpus analysis examines how the possibilities of re-mediation, modification, and composition offered by the flexibility of digital image-video support are exploited.

The corpus was obtained through a manual scan of the profile, highlighting the most explanatory and representative contents. This process was carried out with the utmost attention and precision in order to ensure the accuracy and validity of the results obtained. Segments of text that best reflected the topics of interest were identified in order to provide a comprehensive and detailed overview of the salient aspects of the phenomenon under consideration. Therefore, special attention was paid to the selection of content, prioritizing those that could provide a significant contribution to the understanding of the phenomenon, from a perspective of scientific rigor and objectivity. In particular, it emerges that an image is not necessarily a photograph and can contain linguistic elements. The analysis shows that 38.9% of the posts examined consist of images or videos from other media sources, used through a process of selection, reuse, and in some cases, modification. In other words, the composition of posts on Instagram is often the result of a series of image and text manipulation activities to create new content from existing ones. This happens as if it were a continuous use of a sort of permanent narrative that unfolds through a constant presence on print, TV, and social networks (Ventura 2019), with the difference that everything develops in a single large environment: the digital one.

For example, 10.4% of the posts consist of short videos extracted from national TV programs, such as news or talk shows, and half of which have been modified by adding fixed or moving text. This laborious activity that involves the dissemination of some content from TV programs via social media is defined as dual screening (Giglietto and Selva 2014). Moreover, when it comes to political issues, these continuous references between TV and social media impact both journalists and politicians and the users/public, generating a hybrid experience in which a sort of mediated interdependence appears that entails a wide range of actions by political institutions, media actors, and citizens in a hybrid space of relationship (Ekman and Widholm 2015).

On the other hand, the use of user-generated images/videos, as well as videos, appears sporadic, but significant. From the analyzed corpus, it emerges that 22% of the analyzed Instagram posts are characterized by the presence of a double re-mediation operation, in which the use of online article previews automatically generates the Facebook post format. In this way, social media platforms rework and propose journalistic content in a reduced format, mainly consisting of the title and an image, presented on Instagram. Since Instagram does not allow its users to insert a link to the full article, this re-mediation operation serves the function of attracting users' attention to the news, but does not allow them to access the original source immediately. Other less frequent forms consist of tweet images and newspaper article excerpts (5%). Even regarding the set of journalistic posts, content comes from national online newspapers for 73.5%, with a prevalence of traditional newspapers (73.1%) over native digital ones (26.9%).

Furthermore, in the context of re-mediation, activity on social networks is configured as a practice of selecting and cutting content. In addition to the most followed national television channels, the analyzed posts include quotes from various newspapers. In this case, we highlight an excerpt taken from an Italian television programme aired on *La7* and from a national newspaper such as *Il Manifesto.*

The wide range of available information sources, along with their equal accessibility, provides politicians with a wide choice of information and perspectives from which they can select what they want to propose to their audience on social networks. This means

that politicians are no longer only mediated by journalists, but they are also able to re-mediate journalists themselves, assuming an opinion leader role. Instead of witnessing disintermediation, there is an observed stratification of mediation, which gives politicians a new power of perspective re-creation of the entire public interest debate. In this context, the written word assumes an important role within the images and videos that make up Instagram posts.

Half of the posts (49.6%) indeed contain text within them. While text is rarely the exclusive element, it tends to integrate into the image, constituting what guides the interpretation of the entire post. In addition to the types already mentioned, an example of this is photographs whose main subject consists of signs and banners, posted or displayed during demonstrations (2.3%). Obviously, another group of image–text is represented by the post-posters referring to television appearances or rallies where Elly Schlein is present. Although at times repetitive, as it represents a significant part of the overall data, we have decided to include it in the analysis, given that we are in what is defined as a permanent election campaign. These image-texts are of considerable relevance in the context of social network analysis as they are structured according to graphic-advertising norms, and constitute a digitization of material that would typically be posted on a bulletin board or distributed on the street. In this case, the use of images and text aims to attract the attention of the public and inform them about the promoted event, exploiting the advertising strategies that characterize visual communication. The use of these techniques, although common in advertising communication, requires specific knowledge of visual language and its rules in order to create an effective and engaging message for the public. In this way, poster-posts represent an example of how advertising techniques and visual language are used to promote the presence of a politician and their public events on social networks.

Although less numerous (10%), the images resulting from the composition of various elements are still significant: photographs and screenshots given by elements remediated from tweets, other social posts, or newspapers. Their reading begins with the most prominent linguistic elements, which capture attention and direct interpretation, and then flows towards the next levels where meaning is confirmed and tested. In this case, the vast material made available by the web is exploited to be combined: by juxtaposing sign elements of different natures, internally coherent texts are created, capable of presenting themselves as brief arguments; and, therefore, quickly arousing the inferences and desired conclusions in their users. The condition of usability of these texts, however, consists of the prior knowledge or sharing of the ideological and value vision of the politician. It should also be noted that only a limited portion of the posts (10.3% of the total) consist of images or videos without text that rely on the caption to convey their overall meaning.

As we have seen, there are many cases in which the written word assumes a significant portion of the post's meaning, but relatively few cases in which the verbal component that introduces new information is that of the caption. The caption mostly serves as an accompaniment to the image/video, repeating the keywords, referring to another media, or inviting comments through a question. The role of the caption is to consolidate and clarify the meanings already implicit in the image or video. The image or video should, therefore, have a degree of autonomy from the caption, so that the time required to understand the entire post is minimized. In other words, the caption should serve a supportive and integrative function with respect to the image or video, instead of being the central element of communication. It is no coincidence that the same newspaper articles that are reported are systematically reduced to the form of 'title + image', and are able to exhaust the entire news in the space of the image and with little content. Hashtags, on the other hand, are particularly used in the caption to recall their slogan #partedanoi, which has formed around the topic and leitmotif of the moment (Bentivegna and Boccia Artieri 2019), which can be traced back to the election period. What has been seen fully reflects in multimodality, which refers to the ability of texts to use different semiotic modes, such as verbal language, images, sounds, and other visual or audiovisual elements, to communicate meanings. In this context, modes refer to the means through which meanings can be conveyed and

interpreted. According to this perspective, traditional writing, which uses only verbal language, is a form of unimodal communication, whereas multimodality occurs when multiple semiotic modes are used in a single text. The use of multiple modes, as in this case, can enhance text comprehension and offer new possibilities for representing and communicating meanings. Moreover, authors such as Kress and Van Leeuwen (1996) argue that multimodality has become increasingly common in the digital era, where technology allows for the integration of different semiotic modes into a single form of expression. It is, therefore, a fundamental concept for understanding contemporary texts and their ability to communicate meanings in ever more complex and sophisticated ways.

## 5. Aspiring Secretary & Feminist Champion: Elly Schlein's Daily Life around Italy. Social Media Posts Serve Three Fundamental Functions from a Content Perspective

The first function is to provide an account of the politician's daily activities, with the goal of making their work and priorities known to voters and supporters. The second function is to select news that the politician considers to be most important or relevant to their political agenda, in order to raise awareness of the issues they deem crucial for the community. Finally, the third function is to reference statements made by the politician or others, in order to make known the positions and ideas they support, as well as the opinions of other notable individuals in the political debate. In summary, the functions of political posts are oriented towards communicating the commitments and daily activities of the politician, promoting the relevant themes and issues for their political agenda, and disseminating the statements and positions supported by the politician and others.

Each communicative function is relative to the two different faces of the @ellyesse account, in which both the aspiring leader of a political party and the feminist champion seeking to mobilize a specific part of the electorate, including the young target audience, are defined within the continuity of the same media space and through the same format. The composition, decomposition, and overlapping of these two identities constitute one of the elements that strongly emerge from the analyzed corpus. The interplay between the two figures is clearly evident in the posts that describe daily activities. These posts account for 37.4% of the total, divided almost equally between posts that describe election rallies (36.4%) and posts that photograph mobilization on specific aforementioned issues (43.6%). Another small group of images/videos show Elly as a participant in television appearances (23%). The tour throughout the country is presented either through traditional images of her speaking at rallies or together with party members and activists. These posts are intertwined with those that describe election activity, consisting of photographs taken of the crowd during rallies. In videos, cheers are shown as her entrance on stage is announced or during a specific part of her speech. In this case, the television model remains strongly present in determining the social media format, to the point that the politician is entirely equated with a celebrity. The only difference is that the point of view shifts, either becoming internal to the scene or backstage, which always refers to the entertainment world scenario. It should be noted that the lifestyle-related posts, in clear contrast with what usually happens, are few, but very impactful.

In particular, the photo denouncing the theft of the backpack on the train had a considerable media response, despite venturing into slippery terrain such as security, a strong theme of the center-right in Italy. However, in general, the use of Instagram for political purposes by the new secretary during the primary campaign has been analyzed, and it has been found that it was little used to define herself through her daily life and consumption habits, as often happens for the common user of the platform. This made it difficult to find posts that could be assimilated to those of a common user of the platform. According to the research, it emerged that it was difficult to find social media posts that could be assimilated to those of a common platform user. However, in posts that fell into this category, the importance, suggested by the literature, of political leaders demonstrating values such as coherence, integrity, genuineness, congruence, honesty, reliability, trustworthiness, intimacy, amateurism, spontaneity, and truthfulness (Luebke 2021) was confirmed. The exhibition of such qualities

by a political leader contributes to creating an authentic image that is appreciated by the population and their supporters. In other words, the analysis showed that the use of Instagram by the new secretary during the primary campaign focused mainly on communicating political messages and promoting the political agenda, rather than on sharing personal experiences and creating an empathic bond with the public. This may be due to the historical moment in which the campaign took place, with a strong disaffection and downgrading towards political parties (Dalton and Wattenberg 2003), and the need to show a very controlled public image oriented towards specific purposes. Referring to Goffman's studies, it is certainly another representation of the private that, as such, also involves a backstage. Identity is not a stable and coherent entity of the individual, but rather a construction derived from social interaction and self-representation in different contexts. It is influenced by how he or she represents oneself to others in different social situations. Goffman introduced the concept of performance to describe how individuals represent themselves to others, using masks and social roles based on the context in which they find themselves. Every social interaction is a kind of theatrical performance in which individuals seek to present an image of themselves that is consistent with the expectations of others. However, it shows the peculiarity of fitting into the same routine staged by common users of the platform, so much so that among its followers there is a sense of familiarity and probably solidarity, and this could scratch the unassailable political spectacle. In this case, we witness a politician who neither adopts the media format to which the citizen is accustomed as a mere spectator, nor does he put into practice the same communicative habits of the citizens he wants to address. At the same time, he tries to translate politics into the language of the common citizen, not by speaking the language of the common citizen, but rather by speaking exactly the same language as his audience, in order to unite politician and voters in the common role of social prosumers. The adoption of a form of communication that highlights the normality of the political figure has a dual political significance. First, this communicative choice represents a strong tool of representativeness, as the politician is presented as similar to the public he is addressing, with the aim of creating a more emotional connection with users, generating a perception of closeness (Enli 2015). Moreover, the politician's approach to normality makes it possible to politicize almost every element of everyday life, implicitly using them to convey opinions consistent with one's own ideological perspective, thanks to the overlap between the figure of person and that of politician. After all, current politics is increasingly looking for forms and figures, gestures, and bodies, and not only laws and decrees (Salmon 2014); and this is also due to the Baumanian liquid society (Bauman 2013), in which we are immersed and the electoral volatility (Norris 2011) of which we are protagonists. Unlike pop politics, which adopts a depoliticization strategy to reach audiences who are not interested in politics, in this case we observe an opposite approach, that is, a systematic politicization of elements that only marginally fall within the political sphere. In other words, the political figure uses communication that aims to politicize everyday life and normalcy, to promote their ideological perspective and create a bond of empathy with the audience. This strategy differs from the approach of pop politics, which tries to adapt to the tastes and preferences of the politically inactive audience, instead of trying to politicize the non-political. This aspect is highlighted only when the endorsement of a television celebrity such as Claudio Amendola and, especially, the greeting of the singer Levante at the most popular pop event in Italy, the Sanremo Festival, is emphasized.

## 6. Conclusions

As emerged from the analysis, Elly Schlein's social communication invites us to rethink how the mediatization of politics is currently being approached. On the other hand, another fundamental angle to consider is the consecration, for the first time in the history of the party, of female leadership. The issue of female leadership (Campus 2013; De Blasio and Sorice 2014) is increasingly being used in public debate by parties to position themselves on the issue of gender equality and is, therefore, of considerable interest for the analysis of electoral and digital communication.

It seems to mostly exploit codes and modalities of information structuring already matured within mass media formats, but modifying their overall meaning. On the one hand, the technical modalities through which they are put into practice change: the personal ability of the politician to present themselves as a media personality is accompanied by the work of a team capable of seeking, selecting, and effectively composing the news made available by the hybrid media system. On the other hand, the purpose for which they are used changes because the simplification that results does not serve to entertain an audience only marginally interested in political issues, but rather cements its audience around a narrow set of themes, a community of individuals already to some extent interested in politics. On social media, in fact, the audience cannot help but be made up of those who make politics a defining characteristic of their individuality. The politician, thus, models their communication on the identity-defining logic of these platforms, allowing all the elements that make them up, personally and politically, to coexist alongside each other. It is precisely to achieve this goal that they can also exploit, redefining them, the forms of mass media mediatization. Thus, having established a mimetic and direct link between themselves and their followers, the politician does not just show their own image, but reveals that of the entire public debate. Elly, as a politician and opinion leader, does not only tell her own story, but by doubling the mediation, also tells the story of others about herself and current events. This is where the greatest paradox of social communication lies, which sees, on the one hand, a seemingly entirely projected comparison with all the actors of the hybrid media system, on which it seeks to influence; however, on the other hand, it effectively achieves such openness only on the basis of a prior hermetic closure, in which the relationship with the other and with reality is rigidly exhausted in the always identical and vertical terms of their ideological proposal.

**Funding:** This research received no external funding.

**Conflicts of Interest:** The authors declare no conflict of interest.

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
