# Peer review of "Winning against All Odds: Elly Schlein’s Successful Election Campaign and Instagram Communication Strategies"

_socsci, doi:10.3390/socsci12060313_

Round 1

Reviewer 1 Report

1) This article presents a timely, interesting, necessary and current analysis. Contribution: a) the debate in digital media and social media; b) incorporate various variables to be considered in political communication; c) a case study on the use of Instagram in political communication. It provides valid documentation and various lines of future research.

2) Various investigations are cited in the article. It would be appropriate to have a brief context of the case: “Ellyesse”. Another weakness is the absence of a systematic review of up-to-date research on "social networks and political communication", "social networks and elections" and "social networks and public opinion". It would be convenient for the author to include this type of research. Expand the methodology used. Specify the research questions. Provide the description of the categories and variables analyzed. Provide the procedure for the selection of the sample.

3) The topic is relevant. It would be opportune to expand the bibliographical references with more recent studies.

4) There are cited references. There are few references cited to scientific research articles that delve into the topic of the article in the current use of social networks in political communication. It can be improved.

5) The methodology needs to be expanded. This point can be improved. Contribute: criteria for the selection of the sample. An analysis period is indicated: what are the selection criteria? What are the results obtained from the sample?

6) It would be helpful to provide tables or graphs of the results obtained from the analysis of the sample.

7) The weaknesses in the methodology and in the presentation of the results lead to conclusions that are not deepened.

8) Other specific details in the attached document. Comments in the .PDF file

Author Response

In the article in question, numerous research studies were cited in order to contextualize the case of the new party secretary. Thanks to the feedback received, it was possible to enrich the debate on the topic being discussed. Furthermore, particular attention was paid to the methodology adopted for the realization of the article, in order to ensure its precision and completeness. In fact, an accurate description of the reference context was sought, as well as a detailed analysis of the different sources used for the drafting of the article. In this way, the aim was to offer a scientifically correct and rigorous treatment of the topic at hand.

Reviewer 2 Report

This is an original and cutting-edge study which illustrates a new trend of political mediatization on social media. The theoretical background is solid. Here's a few suggestions to improve the quality of the overall article:

1) The term "remediation" in the title is confusing (I initially thought it meant "correction") and it is only used once in the abstract. It appears hyphenated throughout the article. Since it's a central concept in the analysis, it would be useful to provide a definition at the beginning of the article.

2) Elly Schlein is still a relatively unknown public figure internationally. It would be useful to add a section (perhaps after the introduction or before the analysis) that provides a background of her personal life, her political career, as well as of her social media accounts.

3) The author states on p. 2 that "The second part of the work will focus on the empirical-quantitative analysis." The analysis is predominantly qualitative. It would be beneficial to ground it theoretically in multimodality. I suggest referring to Kress and Van Leuween's work on multimodality (e.g., Reading Images: The Grammar of Visual Design, 2nd ed., Routledge 2006).

4) P. 8. Describe figure 1 in more detail for clarity. It took me a minute to spot the LA7 television channel logo on the left panel. Explain that this is a screenshot of a TV show  which airs on the LA7 national channel.

5) p. 8. Last paragraph. Ground this qualitative multimodal analysis in the literature (see point 3)

6) p. 9. The "Bentivegna & Boccia Artieri" citation is missing the year.

7) P. 9. The author writes "This is not Elly Schlein telling her story in first person, but rather the char-acter of Elly Schlein being mediated by her staff." One could argue that the politician's social media content is also likely to be mediated by her staff.

8) Figure 2. Like for figure 1, the author should provide more contextual information, otherwise the analysis is hard to follow. What is the story behind the Christmas socks and the stolen backpack? Why was this post successful? How is success measured?

9) p. 10. "Referring to Goffman's categories..." Please provide Goffman's citation.

10) The very last paragraph, starting with "This is where the greatest paradox of social communication lies" is not clear and needs rewording. 

Overall, this is a very interesting study. To improve the analysis and the argumentation, I suggest grounding the analysis in the methodological literature on multimodality, adding contextual information to the examples, as well as more information about politician’s background.

Author Response

Thank you

Round 2

Reviewer 1 Report

Thanks for the new version of the article